# Development of an Integrated Water Resource Scheduling Model Based on Platform Plug-In: A Case Study of the Wudu Diversion and Irrigation Area, China

**Beihan Jiang [1], Long Pan [1], Genquan Qin [2], Xiaolin Su [1,2], Feng Cai [1,*] and Yue Liang [3]**

1    College of Civil Engineering, Fuzhou University, Fuzhou 350116, China
2    IStrong Technology Co., Ltd., Fuzhou 350100, China
3    Fujian Key Laboratory of Hydrodynamics and Hydraulic Engineering, Fuzhou 350001, China
*    Correspondence: caifeng@fzu.edu.cn

**Abstract:** Integrated water resource scheduling is a key strategy for controlling river floods as well as for promoting the benefits and abolishing the harmful aspects of water conservancy projects. It is necessary to explore an effective development mode to address the current issues of long development times and poor outcomes for integrated water resource scheduling models. Drawing on the concept of software reuse, a development mode for an integrated water resource scheduling model is offered based on "platform system + model plug-in", the cores of which are plug-in modules and interface integration. The boundaries and connection relations of each plug-in module are formed based on the logical analysis of the model plug-in. A web application mode is used to implement a standardized interface, which can be quickly and seamlessly connected to the system platform. The model is explored and applied in the Wudu diversion and irrigation area in China. The generated model is eventually verified using data obtained from two flood periods. According to the simulation results, the gate operation will be convenient, and the target water level will be attained in the allocated time with a satisfactory peak-clipping effect. It shows both good coordination and great utilization value of the plug-in modules. The concept of designing a plug-in database is eventually taken into consideration for combining and generating an optimization model of each river.

**Keywords:** model plug-in; integrated water resource scheduling; model application platform; interface encapsulation; Wudu diversion and irrigation area

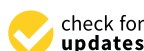



## 1. Introduction

Integrated water resources scheduling is a control technology that formulates water supply strategies under the premise of ensuring the safety of water conservancy projects, based on the operation planning of those projects, and aiming to supply sufficient water resources as much as possible [1,2]. Comprehensive and efficient management of water resources with flood control, drought resistance, and irrigation are made possible by integrated water resource scheduling, which also plays a key role in ensuring water security, water distribution, and water ecology [3–5]. At present, the model and system of integrated water resources scheduling is usually developed for specific areas, and most of them are only for a single function, with poor universality and expansibility [6,7]. In addition, based on the sequence of the water cycle, a sequential development mode is adopted from the hydrological model to a hydraulic engineering scheduling model [8,9]. This mode is hindered by issues in practical application, including lengthy development times and unpredictable results [10,11]. The earlier models and systems of integrated water resource scheduling were easy to upgrade when new functions were demanded due to the relatively simple structure. However, with the increase of the complexity of the model application platform, the upgrading speed is also gradually accelerated, which makes the traditional

model application platform development methods difficult to keep up with in term of the requirements of the current day.

In order to solve the defects of the traditional development mode, improve the development efficiency of the water resources integrated scheduling system, and promote the reusability and interoperation of software, some ideas of software engineering design must be adopted for reference. Plug-in technology is used by many sectors, such as simulation design environments and remote sensing monitoring systems. It is well-known and frequently utilized due to its simple structure and strong secondary growth potential [12,13]. The development of an integrated water resources scheduling model is a long process, and the construction of a model application platform is also a complex project. One of the major challenges academics are now facing is determining how to include plug-in technology into a water resource dispatching model.

With the development of hydraulic informatization, water resource information systems are no longer primarily for data summary and inquiry, and intelligent applications with numerical models as the core have become indispensable parts of the system [14,15]. So far, the stand-alone mode, the Web mode (B/S mode), and the cloud platform mode have been the three major application modes for hydraulic numerical models.

The stand-alone software of the hydraulic numerical model was created first and made significant progress, such as the MIKE SHE and MIKE BASIN series, the Hydrologic Engineering Center-Hydrologic Modeling System (HEC-HMS), and the Storm Water Management Model (SWMM) [16–18]. With the development of Web technology, since 1997, the second generation of network software based on B/S (browser/server) has been popular because of its simplicity and interactivity. The model is often combined with the hydraulic system in the Web application mode, and the calculation is carried out through the system platform on the browser without installing the model software locally [19,20]. The hardware and software resources are integrated and managed in a cloud platform to provide computing, network, storage, and security services in the form of services [21]. Scholars have studied the cloud platform of hydraulic engineering in many fields [22–25]. The cloud platform mode is an advanced form under the B/S mode with the characteristics of fast computing for multiple users and multiple schemes. However, further research is essential because it is also constrained by network security and network speed [26].

The integrated application system has been the subject of a great deal of research over the years. With the aim of system management and from a hydrological, hydrologic, and sediment analytic standpoint, Peng et al. [27] explored the elements and organization of integrated water resource scheduling and management systems in a river basin. Xiao et al. [28], who also carried out system verification, examined the integrated automatic information management system for hydraulic engineering in terms of its functional makeup and software architecture. The core components of the overall software system for allocating water resources are the special database, model function application, model administration, and model calculation service [29]. The technological framework and method of integrated water resource scheduling proposed by Lei et al. [30] are based on the process of human and natural water cycles. Based on the existing circumstances and functional needs of the Manas River in Sri Lanka, the software created by Gan et al. [31] enhanced the overall software architectural layers for integrated water resource scheduling.

In conventional system development, tactics including linear progression, iteration, feedback, and experimentation are commonly employed [32,33]. To improve the efficacy of system integration, a complete integration technique comprising an integration platform, component service, and integrated services as part of the core was suggested by Ma, Z.H. et al. [34]. In the two common development strategies of the integrated water resources scheduling platform, the top-down approach with the platform as the core produces a comprehensive but tedious system, while the bottom-up approach with the model as the core is more detailed but less comprehensive. Thus, the development method of a platform plug-in that integrates the two strategies has good prospective applications.

Researchers constantly seek ways to boost the efficiency of platform development and reduce the repetition of software. A component-based development strategy is proposed. Based on the manner of "integrated framework + model component", a quick generation framework for a model-stabilized water computing system was proposed and implemented by Shi et al. [35]. The development criteria for the components' developments for flood prediction were examined by Zhang et al. [36], and the outcomes demonstrated that component-based development offers several benefits. To reduce the difficulty of creating a model application system in SOA (Service-Oriented Architecture), scholars have developed model components in the form of Web services [37].

There has been less research on the division of components of scheduling models, whereas more research has been conducted on hydrological models such as the Xinanjiang model. Model plug-in development is similar to componentized development. The main difference is the design philosophy. Componentized development divides model modules to promote reusability, which allows for the updated replacement of individual model modules, while plug-in inserts various plug-ins into the program so as to enhance system scalability and flexibility [38]. In southern Italy, Andrea, S. et al. [39] studied a multi-reservoir engineering project. They compared the simulations created by several models running under various operational circumstances and discussed the implementation and evaluation of the model plug-in. There are now few plug-in designs for specific models. As a result, further research has to be done on the division and development of model plug-ins on integrated water resource scheduling models.

In this paper, a new method is proposed for the development of an integrated water resources scheduling platform with a plug-in model as the core. In the "platform system + model plug-in" development mode, the functional plug-ins are created; the task assignment, boundary conditions, and connection relationships between plug-ins are clarified; and the plug-ins are encapsulated and connected to various platforms through the web application mode. Finally, the construction and application of the model are carried out in the Wudu water irrigation area to implement the efficient development of the integrated water resources scheduling system.

## 2. Study Area and Implementation of Model Plug-In

### 2.1. Overview of Wudu Diversion and Irrigation Area

2.1.1. Physical Geography

The Wudu diversion and irrigation area is located in Sichuan Province, China, connecting to the Tanjiazui Reservoir Irrigation Area in the southeast, the Sheng Zhong reservoir irrigation area in the northeast, the Longmen Mountain and Jianmen Mountain foothills in the northwest, and bounded by the Tongkou River and Fujiang River in the southwest. Its 6833 km$^2$ territory is divided into 170 townships and 8 counties (cities and districts). The Wudu diversion and irrigation area is being managed in two parts. By the end of 2000, the first stage had been most completed with a water intake hub, a main canal, and the Fuzi canal, with an irrigated area of 846.53 km$^2$. The Direct Irrigation Area of Wudu Reservoir and the Xizi Irrigation Area make up the second stage, which involves six counties (cities) and an irrigation area of 676.87 km$^2$.

The Fujiang River, which originates in Sansheyi at the foot of Minshan Mountain, is a first-level branch on the right bank of the Jialing River. It travels from the northeast to the northwest. The river basin is 36,272 km$^2$ in size, and the main canal is 675 km long. The relative elevation difference surpasses 5000 m from Snow Treasure Summit (elevation 5588 m), which is the watershed from the Minjiang River to the estuary (approximately 200 m). An image of the specific location from Wudu Reservoir to the intake complex in Mianyang City, Sichuan Province, China, is displayed in Figure 1.

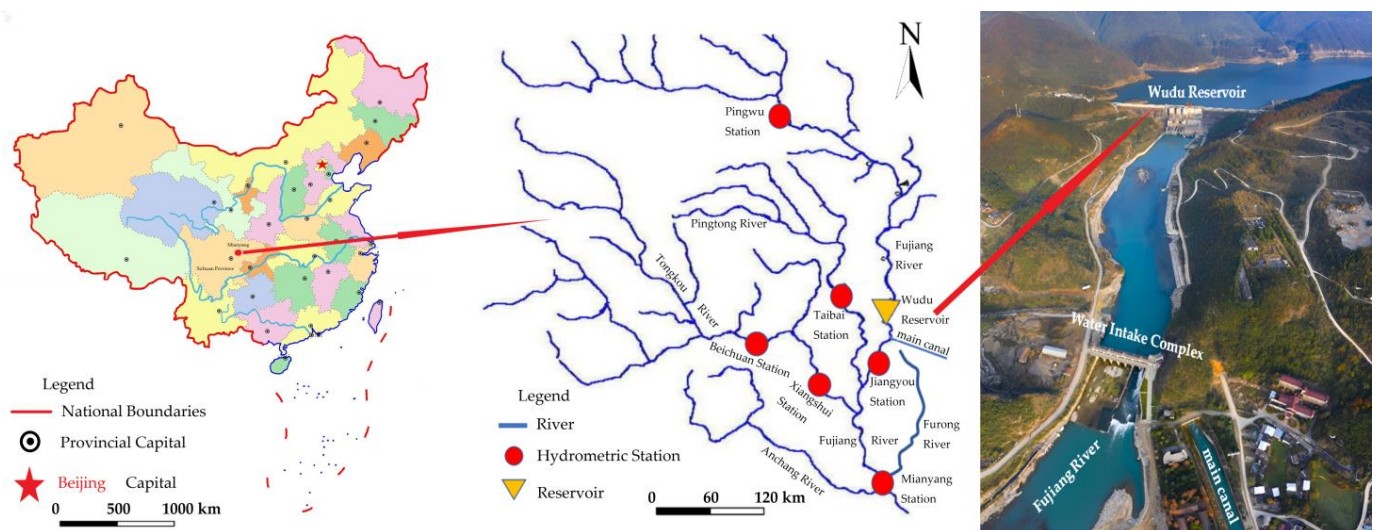

**Figure 1.** Specific location from Wudu Reservoir to the intake complex in Mianyang City, Sichuan Province, China.

The dam site for the Wudu Reservoir is located along the section of the Fujiang River trunk from Moyindong to Maidiwan. The average annual flow at the dam site is 140 m$^3$/s. Most runoff results from rainfall, with small amounts coming from snowmelt, ice water, and groundwater recharge. The reservoir is situated in the Longmen Mountains' concentrated downpour region, with heavy rainfall of high intensity, rapid confluence, and high flow rates. Floods are caused by rainstorms with high peak volumes, which are most common from June to September and are concentrated in the months of July and August. The major flood season lasts from June to September, with transitional periods in May and October, with the dry season lasting from November to April of the following year.

### 2.1.2. Irrigation Business Objects

One water supply reservoir (Wudu Reservoir), three regulating reservoirs on the main canals, one power station, and two water distribution hubs have been the leading business objectives of the Wudu diversion and irrigation area. The distribution relationship between the main canals and their upper nodes is presented in Figure 2.

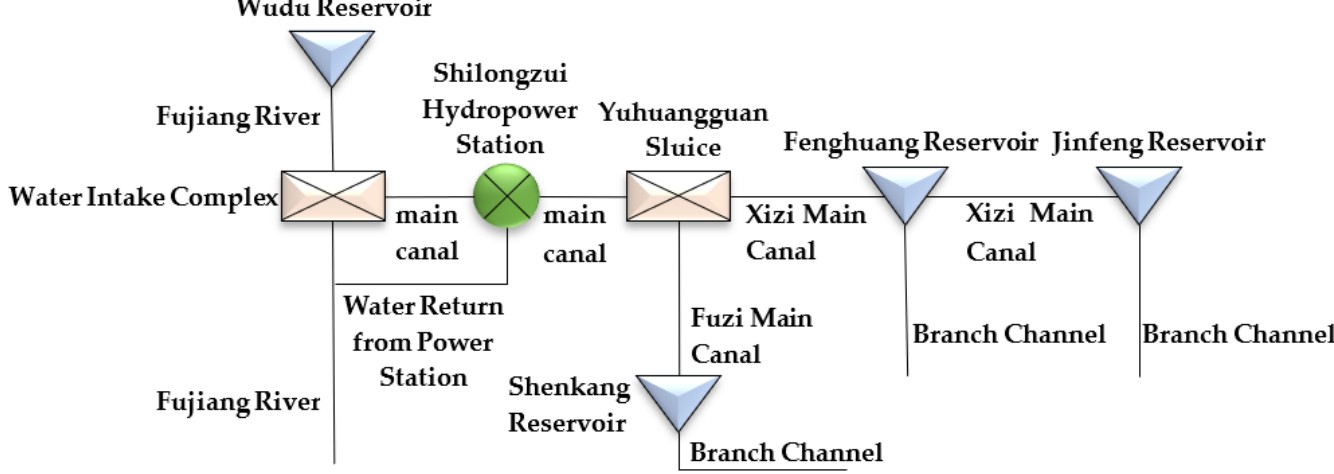

**Figure 2.** An overview of main canals and connections in the Wudu diversion and irrigation area.

For irrigation area systems, the control range and importance of hydraulic structures decrease from the water source in the irrigation area to the water user terminal, so they can

be developed in the sequence from the water source in the irrigation area to the water user terminal. The Wudu water diversion and irrigation area may be split into three categories: reservoir, sluice, and canal system. The Wudu Reservoir and the downstream intake hub are the primary hydraulic structures engaged in the dispatching function for flood control. The dispatching models employed represent the entire irrigation area, and these two hydraulic structures serve as essential dispatching controls for ecological dispatching, power generation, and water supply. Therefore, the development of the basic scheduling model for these two hydraulic engineering structures is particularly important and must be given priority.

*2.2. Overall Design of Integrated Water Resource Scheduling Model Development in the Irrigation Area*

2.2.1. Development and Design of Model Function

The complete process of integrated water resource scheduling includes input and demand water forecasting, water regulation, and hydraulic structure management. The creation of corresponding models is a protracted and challenging task. Hence, the whole integrated water resource scheduling process is divided into modules based on business operations and created as separately designed model plug-ins. In addition to directly modeling the necessary business activities grounded on front-end monitoring data, it may also conduct the scheduling function through a combined application.

According to the objects involved in the Wudu diversion and irrigation area and the application needs, the primary business functions that need to be developed for integrated water resource scheduling in this area include flood prevention and early warning, flood control scheduling, drought resistance and early warning, water supply scheduling, hydropower generation scheduling, and water quality ecological scheduling. The six functions are exhibited in Table 1.

**Table 1.** Business functional requirements and model support table of the Wudu diversion and irrigation area.

| Functions | Main Operating Objects * | Model Support |
|---|---|---|
| Flood prevention and early warning | / | Hydrological model |
| Flood control scheduling | Wudu Reservoir and water intake complex | Flood control scheduling model |
| Drought resistance and early warning | / | Water demand forecasting model |
| Water supply scheduling | Wudu Reservoir, water intake complex, channel system, and gate | Water supply scheduling model |
| Hydropower generation scheduling | Wudu Reservoir and Shilongzui Hydropower Station | Hydropower scheduling model |
| Water quality ecological scheduling | Wudu Reservoir, water intake complex, channel system, and gate | Water quality model |

Note(s): * /: no operating object.

Flood prevention and early warning function: The role of estimating future flood danger and taking appropriate early warning actions is known as flood prevention and early warning. It is also required to employ the runoff production and confluence simulation of the hydrological model to understand the flood process correctly.

Flood control scheduling function: The flood control scheduling aim to ensure the safety of the water conservation project and downstream regions by regulating and managing flood storage and release. The pre-discharge systems designed in advance are used for most dispatching. In order to realize automatic scheduling, it is necessary to develop a corresponding scheduling model and provide a scheduling scheme in the form of a stage water level control target or a gate opening scheme.

Drought resistance and early warning function: The drought resistance early warning function predicts regional water demand for the future period. A water demand prediction model for each water user needs to be built to construct water consumption plans.

Water supply scheduling function: The purpose of water supply scheduling is to develop a water supply scheme and an operation scheme for each hydraulic structure while determining the balance of supply and demand based on incoming water and water demand status.

Hydropower generation scheduling function: A hydropower generation scheduling model must be created to offer a reservoir water level operating scheme that fulfills the premise of flood control safety.

Water quality ecological scheduling function: The function of water quality scheduling offers employees an ecological scheduling scheme, which requires the assistance of a water quality model.

### 2.2.2. Phased Model Development Route

The integrated water resource scheduling model of the Wudu diversion and irrigation area is developed in the following four stages, as shown in Figure 3.

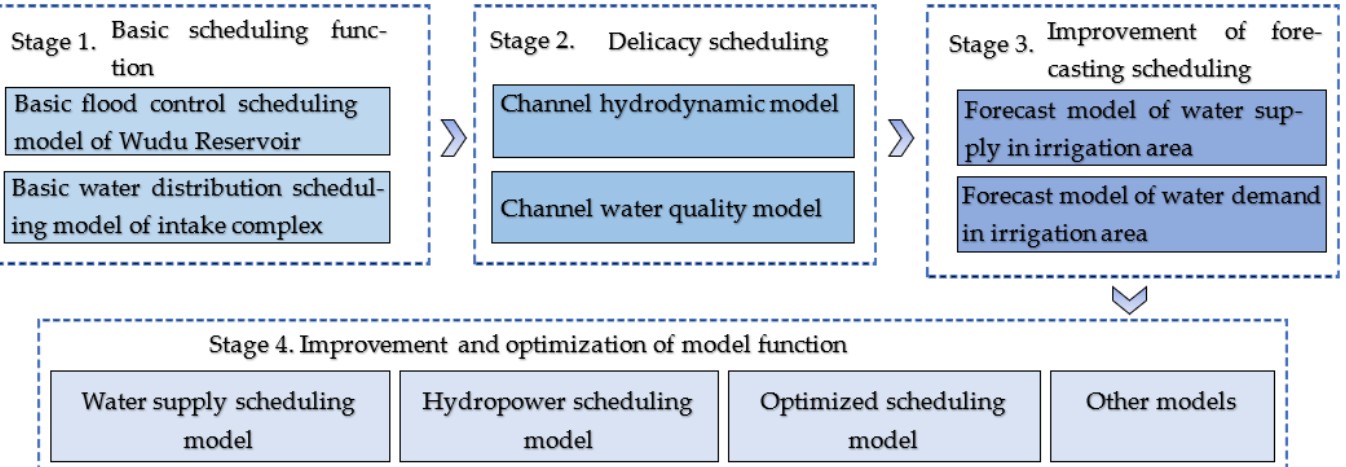

**Figure 3.** Stage-by-stage development design of model functions in the Wudu diversion and irrigation area.

(1)    Stage 1: Basic scheduling function.

The principal water supply for the Wudu diversion and irrigation area, as well as the most critical component of the irrigation area's water regulation, is supplied by the Wudu Reservoir. First, it should be constructed. The Wudu Reservoir serves the dual function of providing water and preventing flooding. The purpose of the flood control schedule is crucial for protecting people's lives and property and acts as a prerequisite and an assurance for subsequent activities. Therefore, the flood control scheduling model must be developed in the first stage of the integrated water resources scheduling application platform.

The fundamental scheduling approach is chosen to make the scheduling rules programmed and modeled, rather than the optimal scheduling method. This is mainly due to the application stage's gradualism and the scheduling function's generality. The application of the new model platform system should be checked and adjusted following a step-by-step process. The system is mainly used as a decision-making tool by the dispatcher. At this time, the initial priority is given to the practical functions. It may be used as a tool all by itself, or it can be implemented in the scheduling function for flood control by fusing many basic operational procedures. Additionally, it enables future portfolio expansion.

The canal headwork sluice portion and the Fujiang sluice portion comprise the intake complex downstream of the Wudu Reservoir. The canal headwork sluice controls the water intake from main canal, while the Fujiang sluice manages flood control, sediment scouring, and ecological dispatching. In addition, it is critical to coordinate with the intake

complex when the Wudu Reservoir releases water for flood management; hence, a related scheduling model of the intake complex should be established.

(2)    Stage 2: Precise scheduling.

A hydrodynamic model of the canal system is created to simulate the flow process, accurately comprehend the flow condition, realize precise water supply scheduling, and also support the water quality simulation in the later stage. Based on the hydrodynamic model, the water quality model is used to simulate dissolved oxygen and contaminants to support ecological scheduling functions such as emergency pollutants dispatching.

(3)    Stage 3: Improvement of forecasting scheduling function.

At this stage, the basic scheduling function of the irrigation area has been finished, and the foreseeable period of the dispatching function should be extended. The forecasting scheduling may be realized by the forecast model of water supply and requirement. The Xin'anjiang hydrological model was applied in the Wudu Reservoir to determine the input flow by simulating rainfall–runoff. The main focus is to develop the corresponding communication interface to connect the current Xin'anjiang model to the application platform system.

(4)    Stage 4: Improvement and optimization of model function.

It is possible to create an optimization model for some dispatching objectives, such as flood control, water supply, hydropower generation, and multi-objective scheduling. The endeavor is now primarily separated into two sections: the first involves improving and expanding the model's functions, and the second involves the upgrading and optimizing of the existing model.

*2.3. Implementation of Core Model Plug-In in Irrigation Area*

2.3.1. Partitioning of Model Plug-In

For simplicity, the developer normally develops a model with the complete function of the flood control scheduling model directly. A disadvantage of this technique is its reliance on the model's accuracy, which significantly influences the dispatching outcome. Errors from various input components are prone to accrue and amplify throughout the operation of model, resulting in an inaccurate simulation. The model cannot perform its role when it is too imprecise because it is difficult to make corrections through human–machine interactions. The efficiency of the optimization algorithm increases the operation time of many scheduling models, especially optimal scheduling models. Numerous elements need to be considered while developing optimal scheduling, which not only directly impact the accuracy of the model, but also significantly lengthen the running time of the model.

A plug-in model technique was proposed to tackle the concerns of single function, impracticality, inflexibility, poor interactivity, and lengthy running time. During model development, the functions of each part of the model are disassembled and packaged into independent plug-in modules, which can be invoked by interfaces and combined to achieve a complete scheduling function.

The functions of the scheduling model are divided in plug-in development. The input and output formats between the models are standardized to facilitate the coupling and recombination between different models. Each part of the model can be used independently, which has more flexibility and practicability so that the staff can make adjustments and have control according to the actual situation.

The function of the plug-in module in the reservoir basic flood control scheduling model is divided into the following seven parts:

(1)    Conversion function between water level and reservoir capacity: The reservoir capacity to query the associated water level and the reservoir capacity to query the corresponding water level can be entered.

(2)  Conversion function between surface and bottom orifice opening and flow rate: Under certain water level conditions, the flow rate and opening of the surface and bottom orifice can be converted from one to the other.

(3)  Calculating the function of the reservoir outflow discharge: Average discharge is determined by combining the forecasted upstream inflow with the current water level and opening under the condition of regulated water level at the end of the specified period.

(4)  Function of gate opening scheme: The average flow rate of the time is converted into the gate opening scheme of the surface and bottom orifice according to the specified opening and closing rules, which consider planned water consumption of the power station, the current water level, and the discharge dispatching rules.

(5)  Function of flood regulating calculation: The process of water level, reservoir capacity, and outflow discharge can be calculated in conjunction with the incoming water forecast and current water level under the specified gate opening scheme. Time periods that do not meet the requirements of the restricted water level or outflow discharge can be screened by specifying the restricted water level or outflow discharge.

(6)  Judgment function of downstream safety discharge restraint: The downstream safety discharge is determined by combining the forecasted downstream inflow and determining whether or not the constraint is fulfilled under the stated pre-discharge flow condition.

(7)  Other functions: At a later stage, more functional models can be created following standard interfaces as needed to increase the number of application scenarios and the level of automated scheduling.

Each part can be used individually, exerting its specific functions, or in combination. The functional design and the interaction between each model component of the basic flood control scheduling model in the Wudu Reservoir are shown in Figure 4.

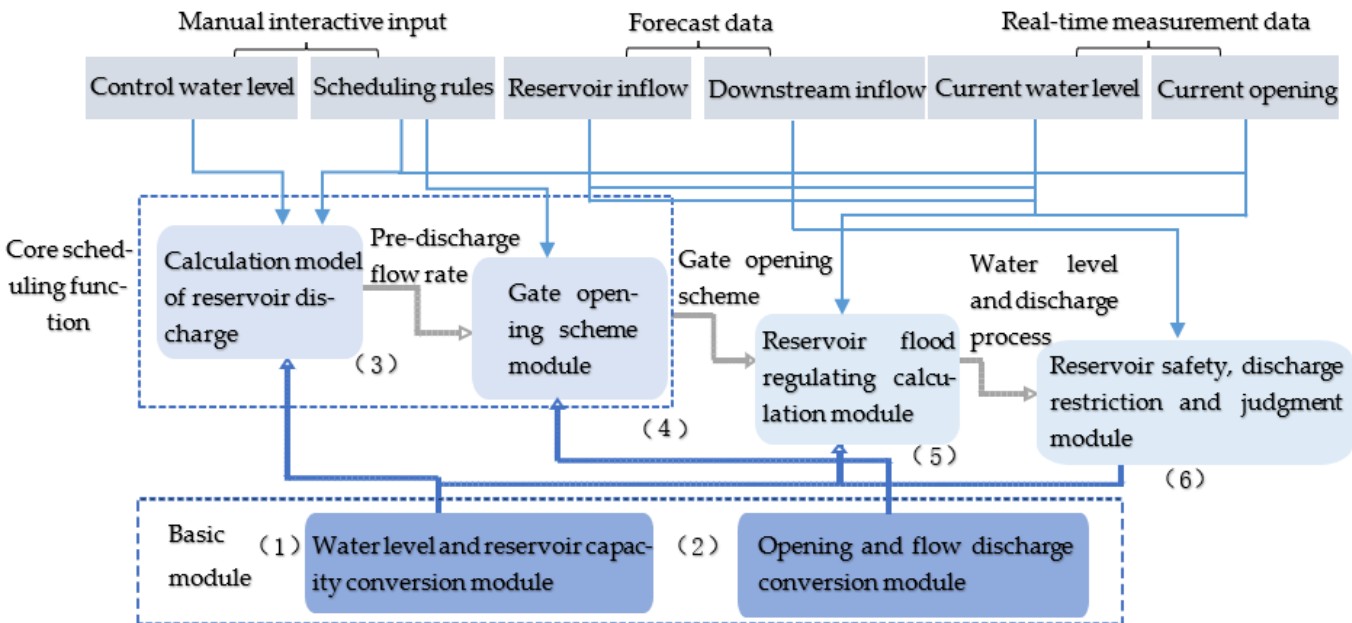

**Figure 4.** Dividing the design of the plug-in module for the reservoir basic flood control scheduling model.

As demonstrated in Figure 4, parts (1) and (2) are basic functions that may be both called independently and implemented as fundamental components of other modules to offer functional support. Parts (2) + (4) can realize the basic operation of flood control dispatching and generate the executable gate opening scheme. Parts (3) + (4) + (5) can further demonstrate the results of flood control dispatching schemes (i.e., water level, reservoir capacity, and outflow discharge process). Parts (5) + (6) are where the gate

opening scheme can be manually specified and adjusted based on the feedback from the constraint judgment.

The function of the plug-in module in the basic water distribution scheduling model of the intake complex is divided into the following four parts:

(1) Function of gate discharge calculation model: The appropriate discharge flow is calculated under the assumption of the stated gate opening combined with the current gate opening, head before and after the gate, gate size, and gate floor elevation.

(2) Function of gate opening calculation: Under the premise of the given discharge, the applicable gate opening scheme is computed using the current gate opening, head before and after the gate, gate size, and floor elevation.

(3) Function of water intake scheme for canal headwork sluice: There is only one sluice gate at the canal headwork section of the intake complex; therefore, the water intake scheme can be directly generated by the gate opening calculation.

(4) Function of water intake scheme for Fujiang sluice: The anticipated discharge of the Fujiang sluice section should be evaluated, and the corresponding gate opening strategy should be estimated while taking ecological, sand scouring, and flood control dispatching laws into consideration.

The functional design and the links between the plug-in modules of the fundamental water distribution scheduling model of the intake complex are shown in Figure 5.

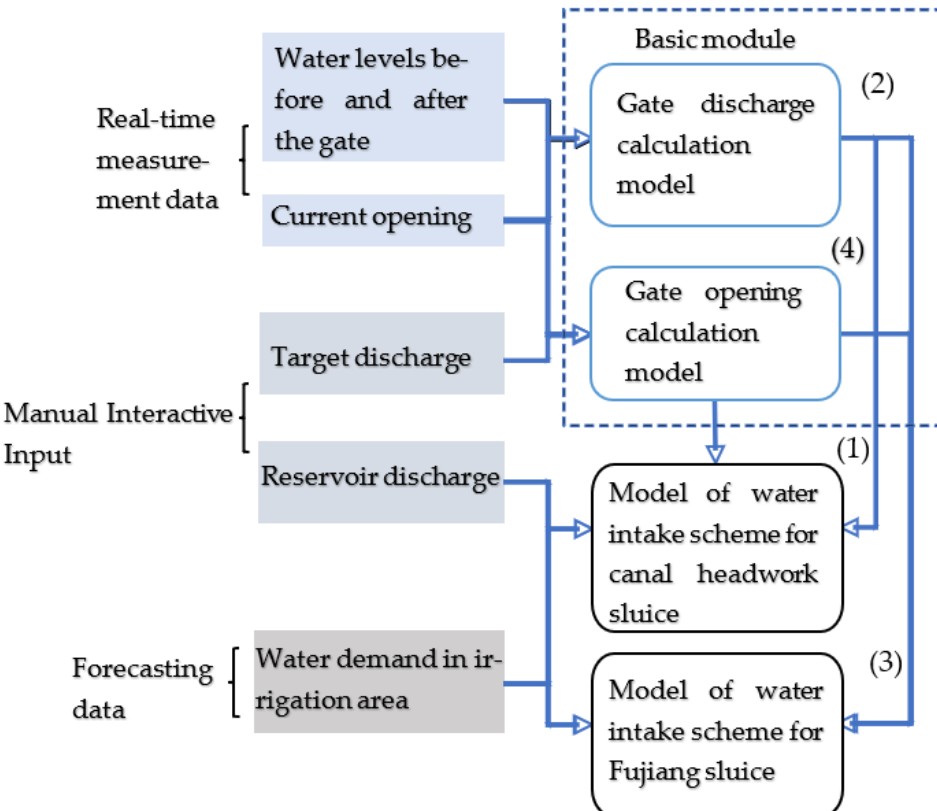

**Figure 5.** Design of the plug-in module division of the basic water distribution scheduling model of the intake complex.

The model plug-ins are developed and designed according to the universal plug-in standard to improve the reusability. The logic division of the reservoir flood control scheduling model plug-in and the basic water distribution dispatching model plug-in of the intake complex is shown in Tables 2 and 3, respectively.

**Table 2.** Plug-in partition of reservoir flood control scheduling model.

| Module Name | Plug-In Name | Input | | Output * |
|---|---|---|---|---|
| | | Model Parameter | Variable * | |
| Reservoir basic attribute module | Converting water level to reservoir capacity | Relationship between water level and reservoir capacity | $Z$ | $W$ |
| | Converting reservoir capacity to water level | Relationship between water level and reservoir capacity | $W$ | $Z$ |
| | Calculation of reservoir outflow discharge | Sluice gate opening, floor elevation, and number of orifices | $Z, e$ | $q$ |
| | Calculation of reservoir gate opening | Sluice gate opening, floor elevation, and number of orifices | $Z, q$ | $e$ |
| Reservoir basic scheduling module | Calculation of average pre-discharge flow rate | $\varepsilon$ | $Qls, Z, ZT, qdz$ | $qxx$ |
| | Gate operation scheme | Discharge dispatching rules | $Z, qxx$ | $e$ |
| | Flood regulating calculation | Relationship between reservoir water level and outflow discharge | $Qls, Z, e$ | $q(t), Z(t)$ |
| | Judgment of safety discharge constraint | Rules for determining safety discharge | $q$ | Judgment results |

Note(s): * $Z$: Water level; $W$: Reservoir capacity; $e$: Gate opening; $q$: Flow rate; $\varepsilon$: permissible error of flow rate; $Qls$: Inflow discharge of reservoir; $ZT$: Goal level; $qdz$: Inflow discharge of hydropower station; $qxx$: Average discharge in a period; $q(t)$ and $Z(t)$ refer to variations of water level and discharge with time.

**Table 3.** Plug-in partition table for the basic water distribution dispatching model of the intake complex.

| Module Name | Plug-In Name | Input | | Output * |
|---|---|---|---|---|
| | | Model Parameter | Variable * | |
| Basic attribute module of intake complex | Discharge calculation of broad-crested weir | Size of weir, elevation of gate, and number of orifices | $hu, hd, e$ | $q$ |
| | Gate opening calculation of broad-crested weir | Size of weir, elevation of gate, and number of orifices | $hu, hd, q$ | $e$ |
| Basic water distribution scheduling module of intake complex | Water intake scheme for canal headwork sluice section | Size of weir, elevation of gate, and number of orifices | $hu, hd, qtar$ | $e\_qs$ |
| | Water intake scheme for Fujiang sluice section | Size of weir, elevation of gate, and number of orifices | $hu, hd, qtar, qxxwd, e0$ | $e\_fj$ |

Note(s): * $hu$: Full head in front of the weir including upstream velocity; $hd$: Water depth behind the gate; $qtar$: Target discharge flow; $qxxwd$: Outflow discharge of Wudu Reservoir; $e0$: Initial opening of sluice orifice; $e\_qs$: Gate opening scheme for canal headwork sluice section; $e\_fj$: Gate opening scheme for Fujiang sluice section.

### 2.3.2. Creation and Encapsulation of Model Plug-In

(1)    Developing languages and tools

Model functions are applied in the form of plug-in services, which are cross-linguistic and cross-platform, as they only relate to interfaces and are not constrained by the development language or platform. Therefore, model plug-ins can be developed in any development language.

The benefits of the Python programming language are its ease of learning, large class library, robust extensibility, and robust function set. It is object-oriented, declarative, universal, and open source. It offers a wide range of potential applications and is suitable to create model plug-ins [40,41]. An application software called the Python IDE provides the integrated development environment (IDE) for Python programs. Code creation, analysis, building, and debugging are all included in the overall development software service package. One of the most widely used Python IDEs nowadays is called PyCharm. It integrates a variety of development tools and has robust and potent capabilities, particularly suitable for Web development, project development, and artificial intelligence, which may

be beneficial for developers. In light of this, PyCharm (Edition 5 February 2020) is used in this research to create a Python-based model program.

(2)　Plug-in implementation

The types and methods of plug-ins for the reservoir and intake complex are shown in Tables 4 and 5, respectively.

**Table 4.** Reservoir plug-in creation.

| Plug-In Name | Classes | Methods | Functions |
|---|---|---|---|
| Converting water level to reservoir capacity | ReservoirAttribute | level_to_volume() | Converting reservoir water level into corresponding reservoir capacity |
| Converting reservoir capacity to water level | ReservoirAttribute | volume_to_level() | Converting reservoir capacity into the corresponding water level |
| Calculation of reservoir outflow discharge | ReservoirAttribute | discharge_calculate() | Calculation of outflow discharge through reservoir water level and gate opening |
| Calculation of reservoir gate opening | ReservoirAttribute | gatage_calculate() | Calculation of gate opening by reservoir water level and outflow discharge |
| Calculation of average pre-discharge flow rate | ReservoirRegulation | average_ pre_ discharge() | Calculating average pre-discharge flow rate in future period with forecast data |
| Gate operation scheme | ReservoirRegulation | sluice_operation_scheme() | Opening and closing scheme sluice gate |
| Flood regulating calculation | ReservoirRegulation | flood_routing() | Flood calculation according to incoming water forecast and discharge plan |
| Judgment of safety discharge constraint | ReservoirRegulation | safety_discharge() | Determining whether outflow discharge conforms to safety discharge restriction |

**Table 5.** Water intake complex plug-in creation.

| Plug-In Name | Classes | Methods | Functions |
|---|---|---|---|
| Discharge calculation of broad-crested weir | BarrageUniversal | broad_crested_weir_discharge() | Calculating the flow rate of the broad-crested weir with a specified gate opening |
| Gate opening calculation of broad-crested weir | BarrageUniversal | broad_crested_weir_gatage() | Calculating the gate opening of the broad-crested weir with a specified flow rate |
| Water intake scheme for canal headwork sluice section | BarrageRegulation | barrage_operation_scheme_qs() | Gate opening scheme of canal headwork sluice section |
| Water intake scheme for Fujiang sluice section | BarrageRegulation | barrage_operation_scheme_fj() | Gate opening scheme of Fujiang sluice section |

(3)　Encapsulation of plug-in interface

The model must be encapsulated since the model plug-in developed in this work is invoked as a service. The details of the model are hidden in the encapsulation, and the model functions are packaged, leaving only one communication interface to external interaction. The strict access interfaces, that is, the communication rules between the model and the outside, must be defined in the encapsulation.

To make the encapsulation rules lightweight, the API (Application Programming Interface) are applied as the model communication interface, the Hypertext Transfer Protocol (http) as the communication protocol, and the light-weight data exchange format (json) to define the data content.

The Flask framework, a lightweight Web application framework created in 2010, was selected as the interface development tool. Flask is a complete communication development framework that defines both communication specifications and implementation technologies [42]. The core of Flask consists of the Jinja2 template engine and the WSGI (Web Server Gateway Interface) from Werkzeug. As a lightweight framework, the application framework of Flask is equivalent to a kernel. Most of its functions are implemented by extension packages, which makes Flask very portable and flexible [43]. Additionally, Flask is widely applied in API interface development because of its security and clarity of sample documents.

The plugins of the integrated water resources management platform proposed in this paper can be developed independently and used as model components. The model invokes the services of plug-ins through explicit interfaces. Developers can easily combine plug-ins into larger programs without worrying about the implementation details, because only the interface information is exposed to the user, which offers a true sense of encapsulation.

## 3. Results and Discussion

### 3.1. Verification of Model Plug-In

The water level and inflow and outflow discharge of the reservoir were collected during two flood periods, from 5:00 on 26 September 2021 to 15:00 on 27 September 2021 (period 1), and from 2:00 to 12:00 on 8 October 2021 (period 2), to validate and evaluate each plug-in module. The measured data are shown in Figures 6 and 7.

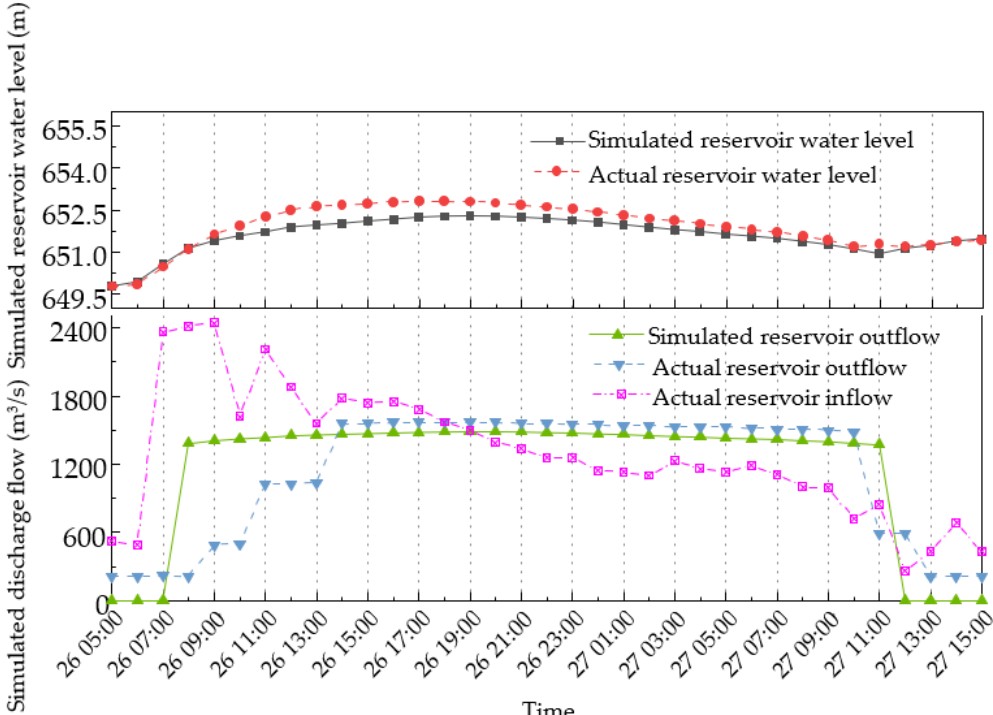

**Figure 6.** Comparison of water level and outflow discharge between simulated results and actual process under Condition A3.

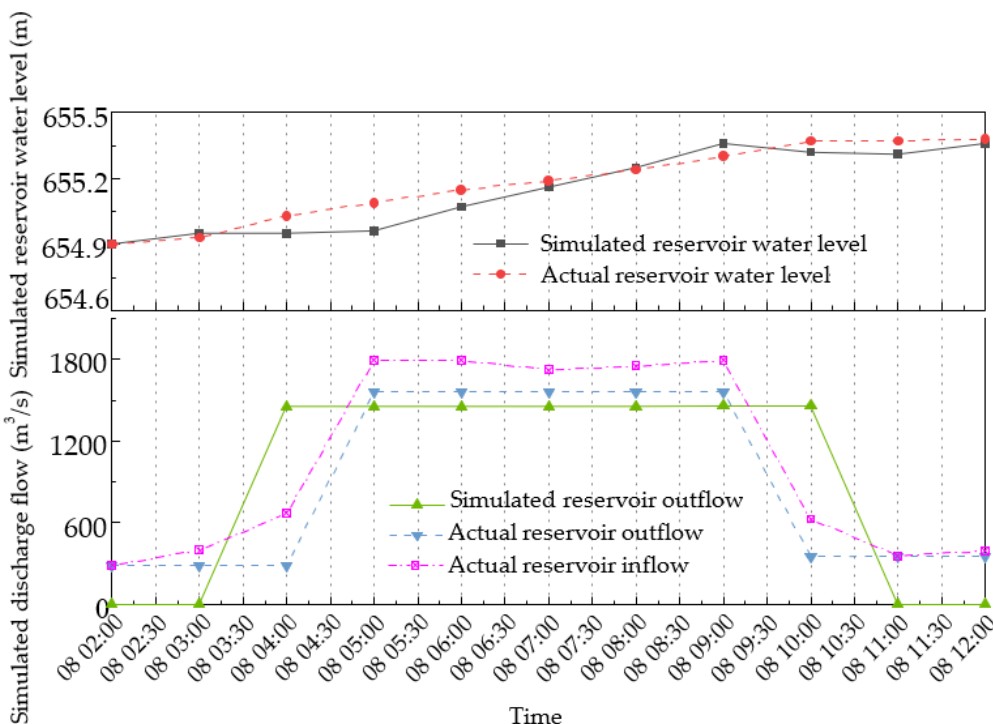

**Figure 7.** Comparison of water level and outflow discharge between simulated results and actual process under Condition B3.

3.1.1. Model Plug-In of Reservoir Outflow Discharge Calculation

(1)    Condition setting

To verify the accuracy of the plug-in of the calculation model for reservoir outflow discharge, the actual discharge process of periods 1 and 2 was taken as a reference to reproduce and validate the operating conditions A1 and B1. Various initial and end water levels were employed as variables, as shown in Table 6. Since there were no downstream safety discharge constraints due to the total measured flow at Taibai Station and at Xiangshui Station after 2 h, being less than 3000 m³/s, the number 0 could be used in the computation.

**Table 6.** Simulated conditions of plug-in model of reservoir outflow discharge calculation.

| Operating Condition | Period | Initial Water Level (m) | End Water Level of Specified Period (m) | Initial Gate Opening (m) | Other Water Consumption * (m³/s) | Flow Rate of Taibai Station (m³/s) | Flow Rate of Xiangshui Station after 2 h (m³/s) |
|---|---|---|---|---|---|---|---|
| A1 | 1 | 649.78 | 651.44 | $e_{0so}$ *= [0, 0]<br>$e_{0so}$ * = [0, 0, 0] | 0 | 0 | 0 |
| B1 | 2 | 654.90 | 655.38 | $e_{0so}$ *= [0, 0]<br>$e_{0so}$ * = [0, 0, 0] | 0 | 0 | 0 |

Note(s): * The subscript *so* indicates the surface orifices' openings; other water consumption refers to the water for hydropower, or diversion directly from the reservoir, etc., rather than the flow from the surface and bottom orifices.

(2)    Simulation results

The results of the plug-in model of the reservoir outflow discharge calculation are shown in Table 7. The average outflow discharge of condition A1 reached 1386 m³/s, while that of condition B1 was 1432 m³/s.

**Table 7.** Simulation results of plug-in model of reservoir outflow discharge calculation.

| Operating Condition | Time | Average Outflow Discharge (m³/s) |
|---|---|---|
| A1 | 26 September 2021 05:00:00~26 September 2021 07:00:00 | 0 |
| | 26 September 2021 08:00:00~27 September 2021 11:00:00 | 1386 |
| | 27 September 2021 12:00:00~27 September 2021 15:00:00 | 0 |
| B1 | 8 October 2021 02:00:00~8 October 2021 05:00:00 | 0 |
| | 8 October 2021 06:00:00~8 October 2021 10:00:00 | 1432 |
| | 8 October 2021 11:00:00~8 October 2021 12:00:00 | 0 |

### 3.1.2. Model Plug-In for Gate Opening Scheme

(1)  Condition setting

The average outflow discharge and initial water level from the above simulation of the reservoir outflow discharge calculation were accepted as the input of the plug-in module for the gate opening scheme. As illustrated in Table 8, the bottom orifice rule was set as a variable for the four operating conditions of A2, B2, C2, and D2.

**Table 8.** Simulated conditions of plug-in model of gate opening scheme in Wudu Reservoir.

| Operating Condition | Average Pre-Discharge Flow Rate (m³/s) | Initial Water Level (m) | Bottom Orifice Rule * |
|---|---|---|---|
| A2 | 1386 | 651.13 | R0 |
| B2 | 1386 | 651.13 | R1 |
| C2 | 1432 | 654.95 | R0 |
| D2 | 1432 | 654.95 | R1 |

Note(s): * Rule R0 indicates that three bottom orifices are opened and closed simultaneously; Rule R1 indicates the middle bottom orifice 2 is opened and closed first, and then the bottom orifices 1 and 3 on both sides are opened and closed.

(2)  Simulation results

The simulated gate opening scheme is shown in Table 9. The average outflow was close to the average pre-discharge flow within a reasonable range. The simulation findings for A2 and B2 were the same, since the two cases merely used the surface orifice for discharging, so they were unrelated to the bottom orifice rule. The average outflow discharges of C2 and D2 were 1454 m³/s and 1418 m³/s, respectively, which were different from the projected average discharge of 1432 m³/s. This was due to discontinuity of the gate opening. Since the minimal change of the gate opening in actual operation is 0.01 m, which restricts the exact correspondence between the gate opening and the discharge rate, only a recommended outflow discharge value closest to the objective could thus be determined.

**Table 9.** Simulation results of plug-in model of gate opening scheme in Wudu Reservoir.

| Operating Condition | Simulated Gate Opening Scheme (m) | Average Outflow Discharge under Simulation Scenario (m³/s) |
|---|---|---|
| A2 | $e_{so}$ = [9.21, 9.21], $e_{bo}$ *= [0, 0, 0] | 1386 |
| B2 | $e_{so}$ = [9.21, 9.21], $e_{bo}$ *= [0, 0, 0] | 1386 |
| C2 | $e_{so}$ = [0, 0], $e_{bo}$ *= [2.6, 2.6, 2.6] | 1454 |
| D2 | $e_{so}$ = [0, 0], $e_{bo}$ *= [0.4, 7, 0.4] | 1418 |

Note(s): * The subscript bo indicates the bottom orifices opening.

### 3.1.3. Plug-In Model of Reservoir Flood Regulating Calculation

(1)  Condition setting

Three alternative simulation results from previous gate opening scheme plug-in module (simulation results of condition A2 and B2 were the same) were taken as the input

variables for this plug-in module. Table 10 illustrates three operating conditions of A3, B3, and C3. The restricted water level was indicated as the artificial safety limitations and could be set to the normal water level of 658 m. The safety discharge was the constraint of the outflow while considering the security of downstream flood control.

**Table 10.** Simulated conditions of plug-in model of reservoir flood regulating calculation.

| Operating Condition | Reservoir Inflow (m³/s) | Initial Water Level (m) | Gate Opening Scheme (m) | Restricted Water Level (m) | Safety Discharge (m³/s) | Other Water Consumption (m³/s) |
|---|---|---|---|---|---|---|
| A3 | Period 1 | 649.78 | $e_{so} = [9.21, 9.21]$, $e_{so} = [0, 0, 0]$ | 658 | Unrestricted | 0 |
| B3 | Period 2 | 654.90 | $e_{so} = [0, 0]$, $e_{so} = [2.6, 2.6, 2.6]$ | 658 | Unrestricted | 0 |
| C3 | Period 2 | 654.90 | $e_{so} = [0, 0]$, $e_{so} = [0.4, 7, 0.4]$ | 658 | Unrestricted | 0 |

(2)    Simulation results

The simulation results of the plug-in module for reservoir flood regulation and calculation are shown in Figures 6–8.

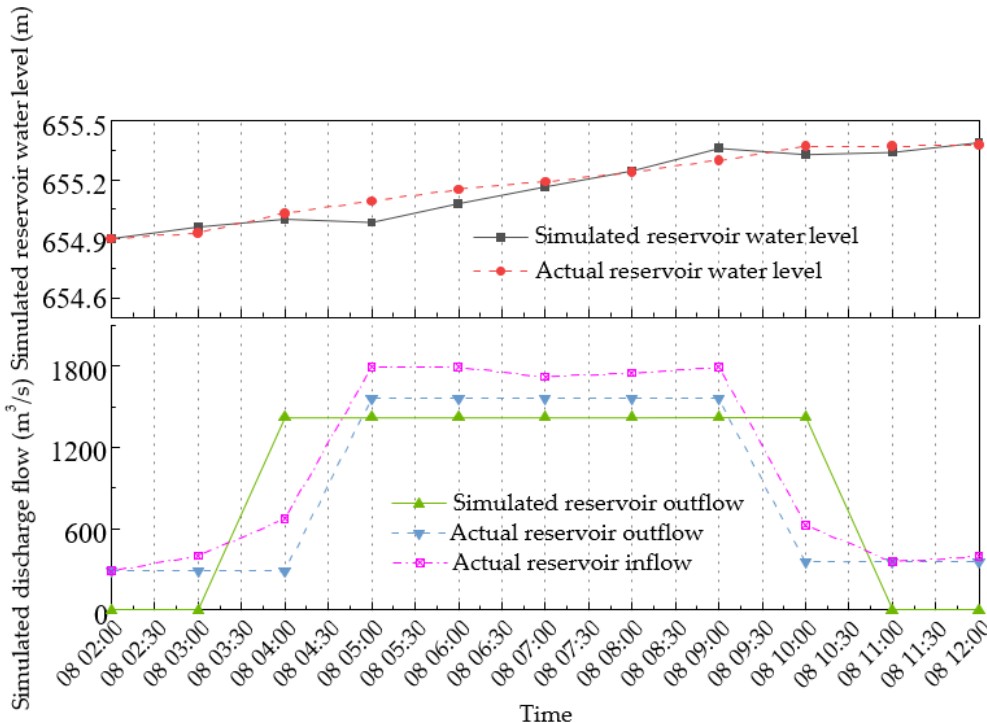

**Figure 8.** Comparison of water level and outflow discharge between simulated results and actual process under Condition C3.

As can be seen from the figures above, the process of the simulated water level was not much different from the actual situation, which basically met the aim of the end-stage water level of the dispatching period. The simulated maximum discharge was smaller than the actual one with a better peak clipping. In addition, the gate adjustments of the simulation scheme were fewer and more convenient for daily operation. Therefore, the feasibility of the flood regulation calculation module was good. Moreover, it showed that the plug-ins functioned well together. The simulation results may meet project requirements with substantial practical significance.

*3.2. Discussion*

The simulated water level in Figure 6 is lower than the actual data for most of the time, because a larger discharge flow is adopted in the model than the actual one at the beginning, resulting in a decline of the simulated water level. In the middle part of the period, the simulated outflow discharge is close to the actual one, so the simulated water level is still persistently lower than the actual data. The simulated initial water level being low is beneficial for flood control but may lead to both a lower water head and a lesser benefit for the power station. In order to improve the practicality of the simulation, the scheduling model can be further optimized. In order to be closer to the actual result, the total scheduling process was divided into several smaller periods, and the restriction conditions of each period were set through manual interaction [44,45]. This is actually from the idea of the mathematical programming method, which can be regarded as a direction of continuous improvement.

The simulated settings for conditions B3 and C3 in Figures 7 and 8 are quite similar. The primary difference is that the end-stage water level in condition B3 (655.36 m) is marginally lower than the objective water level after the dispatching period (655.38 m), while the water level in condition C3 (655.39 m) is slightly higher. This is owing to the various bottom orifice rules for the two conditions, which provide two alternative gate opening schemes when combined with the limitation of the minimum gate opening. As a result, the outflow discharge and water level fluctuate.

Compared with traditional development methods, the plugin-based platform development proposed in this paper offers the following advantages: (1) Transparency of plug-in location. Plug-ins and programs with them can be run in the same process, in different processes, or on different machines. (2) Independence of programming language. Plug-ins are released in binary format, regardless of the language or development environment. (3) Outstanding scalability. Each plug-in is independent and interacts with the outside through its own public interface. A plug-in can extend new requirements by providing new interfaces without affecting the original users, and the new users could also re-select a new interface to obtain the new service. (4) Excellent synergy. Plug-in developers follow a common standard, so plug-ins can be produced, released, and deployed independently from different developers and can be easily incorporated in different applications. (5) Improving the reuse of the system platform. The program code of the original system platform could be packaged into reusable plug-ins through the standard interface, thus protecting the investment of the system platform.

The fundamental dispatching portion of the reservoir and intake complex is divided and designed in this paper on the plug-in creation of integrated water resource scheduling models. Still, the models affected by various dispatching objects are distinct. Therefore, the plug-in partition of other models in the system of water resource scheduling can be explored in future research. As the number of plug-ins increases, we can build a large plug-in database. The database of the integrated water resources scheduling model can be provided by a combination of different plug-ins. The optimal model for a particular watershed can be selected from adequate models of the database.

## 4. Conclusions

This paper mainly focuses on the establishment of an integrated water resources scheduling platform. Learning the idea of reuse in software development, a design scheme with a model plug-in as the core and a "platform system + universal plug-in" as the development mode is suggested. According to the development content and emphases, core and functional plug-ins are created to address various issues existing in the development of the integrated water resources scheduling platform.

The model plug-in development is applied to the Wudu diversion and irrigation area, and seven plug-in modules of the basic scheduling model for the Wudu Reservoir and four plug-in modules of the water distribution scheduling model for water intake engineering are designed. The boundary and connecting relationships of the plug-in are established

based on the logical analysis of the plug-in. The classes and methods of each plug-in module are specified by a universal plug-in, and the plug-in interface is encapsulated through the Web service, allowing the services to be conveniently and quickly invoked.

The water level and flow data from two flood occurrences in 2021 are simulated in the model plug-in. The results show that the desired water level is reached at the end of the dispatching period, and that peak-clipping is more efficient, and gate operation is more convenient. It proves respectable applicability and accuracy of the developed model plug-in. However, it is also noted that the simulated initial water level is lower than the actual data, which is beneficial for flood control but is a disadvantage for the benefits of the power station. Future research on the optimization scheduling scheme of the model is required to increase the viability of the simulation.

Finally, the idea of developing a plug-in database is discussed. Plug-ins for other models can be further developed and designed in terms of creating the optimal model for each watershed and hydraulic engineering situation.

**Author Contributions:** Conceptualization, B.J.; methodology, G.Q.; software, Y.L.; validation, X.S.; investigation, F.C.; writing—review and editing, L.P. All authors have read and agreed to the published version of the manuscript.

**Funding:** This research was funded by National Natural Science Foundation of China (No. 51509038) and National Natural Science Foundation of China (No. 51709048).

**Data Availability Statement:** Not applicable.

**Acknowledgments:** The authors would like to thank IStrong Technology Company and Fujian Key Laboratory of Hydrodynamics and Hydraulic Engineering for their technical support.

**Conflicts of Interest:** The authors declare no conflict of interest.

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
