# Peer review of "Development of an Integrated Water Resource Scheduling Model Based on Platform Plug-In: A Case Study of the Wudu Diversion and Irrigation Area, China"

_water, doi:10.3390/w14223671_

Round 1
Reviewer 1 Report
The paper is quite good and enjoyable to read. It is quite rare to see modern software architecture practices employed in water resources modelling that is often viewed as an 'academic' domain where industry software best-practices are seldom applied.
That being said, there are some oddities with the paper that should be addressed before being accepted for publication. One major issue is the repetitive use of bullet-point (itemized) lists, that in many cases should either be enumerated or re-organized into a normal logical paragraph structure. This is especially true in the conclusions section. I have never seen a final section of a paper written in this manner.
The manuscript should also be checked over for proper English usage and logical sentence flow, preferably by someone fluent in English or with extensive English language skills.
For instance, the paragraph spanning lines 77-84 seems misplaced and possesses an odd sentence structure. How does a paragraph about Java follow from the preceding paragraph? "The initial version [of what? the solution being presented here?] were written in VB, C#, and other languages."
Service Oriented Architecture (SOA) is mentioned here, and the authors propose a plug-in style design. It is perhaps worth mentioning some other modern design patterns and how they might be useful in this kind of modelling. E.g., Client-server applications (built on the TCP/IP communications stack or HTTP/REST/SOAP), or say, a microservices-like component-wise approach.
Additionally, trade-offs of implementing in a modern cross-platform enterprise language like Java, Python or C# should be weighed against a low-level language like Fortran or C/C++. Some academic groups insist that to achieve desirable performance, a low-level language is 'a must' in modelling codes.
Line 155: There is a lone bullet point for 'Development Content'. Should this perhaps be another subsection?
Other bulleted lists: Either change to numbered lists if appropriate, or I would suggest just writing these as logical paragraphs. Certainly, as mentioned above, the Conclusions should be written in paragraph structure.
The figures look okay to me, some oddities with text labels overlapping axis lines which can be fixed up if possible. Otherwise, these are quite good at conveying the key results.
Author Response
Reviewer #1:
- Response to comment:
The paper is quite good and enjoyable to read. It is quite rare to see modern software architecture practices employed in water resources modelling that is often viewed as an 'academic' domain where industry software best-practices are seldom applied.
Response: Thank you for your comments and recognition, we are glad to hear that.
- Response to comment:
That being said, there are some oddities with the paper that should be addressed before being accepted for publication. One major issue is the repetitive use of bullet-point (itemized) lists, that in many cases should either be enumerated or re-organized into a normal logical paragraph structure. This is especially true in the conclusions section. I have never seen a final section of a paper written in this manner.
Response: We are very sorry for our negligence of formatting issues. The journal template allows the use of bullet-point lists, so we use that in the previous manuscripts. We have made correction of bullet-point lists according to your comments.
- Response to comment:
The manuscript should also be checked over for proper English usage and logical sentence flow, preferably by someone fluent in English or with extensive English language skills.
Response: We are sorry for the English expression. The language issues has been modified.
- Response to comment:
For instance, the paragraph spanning lines 77-84 seems misplaced and possesses an odd sentence structure. How does a paragraph about Java follow from the preceding paragraph? "The initial version [of what? the solution being presented here?] were written in VB, C#, and other languages."
Response: It is really true as Reviewer pointed out that lines 77-84 don't connect well with the content before. Since programming language is not the main research object of this paper, this paragraph has been deleted in the revised version, and only an explanation for choosing the Python as the developing language is made in the lines 367-381. For the choice of language, you can also see the response 5.
- Response to comment:
Service Oriented Architecture (SOA) is mentioned here, and the authors propose a plug-in style design. It is perhaps worth mentioning some other modern design patterns and how they might be useful in this kind of modelling. E.g., Client-server applications (built on the TCP/IP communications stack or HTTP/REST/SOAP), or say, a microservices-like component-wise approach.
Response: Thank you for your good suggestions.
The software application server with the C/S structure has a light load, but the data storage is scattered in both the client and the server, so the data cannot be truly shared and unified, resulting in high maintenance costs. Because of the cross-platform languages such as Java, B/S structure is fiercely competing with C/S structure, with the advantages of convenience and low cost. The current B/S structure is also more aligned with the requirements of Web services, and has a better application prospect on the cloud platform.
SOA is different from the two categories mentioned above, where the connection between different functional units (services) is considered. It is a way of application integration. The layered software system may still be B/S structure. The core of SOA is service, which is not only a function, but also can be defined as objects, applications, etc. SOA can be applied to any existing system, and it does not need to deliberately follow a special communication mode.
Micro-service is decentralized on the basis of SOA, and further provides components and services. Its communication is generally based on REST protocol, which is more lightweight than SOA. The development of micro-services is more flexible and faster, which can respond to the changes of requirements and business updates more quickly. It has a strong reference significance in the process of modeling and integration. The model plug-in service modeling proposed in this paper is also based on such patterns and considerations.
The content of application mode of hydraulic numerical model is added in lines 57-75.
- Response to comment:
Additionally, trade-offs of implementing in a modern cross-platform enterprise language like Java, Python or C# should be weighed against a low-level language like Fortran or C/C++. Some academic groups insist that to achieve desirable performance, a low-level language is 'a must' in modelling codes.
Response: As the reviewer pointed out, it is true programming language can be divided into three categories from low level to high level, namely machine language, assembly language and high-level language. There are also "high and low" levels in high-level languages, such as C, which is the underlying language for many languages, including Python. The lower the language is, the closer the form is to the machine instruction, the more efficient the execution is. However, it is more difficult to write with poor portability. The higher the language is, the more consistent it is with human thinking. The code quantity is small with good maintainability and portability, as well as high implementation efficiency.
High-level languages need to be translated into machine code, so redundancy is possible with low efficiency. The choice of language needs to be balanced. But as computer performance increases, the impact of operational efficiency can be reduced, especially for smaller programming. For hydraulic models, the amount of programming is not too exaggerated. Therefore, in terms of development efficiency and cross-platform application, it is natural to prefer high-level languages.
- Response to comment:
Line 155: There is a lone bullet point for 'Development Content'. Should this perhaps be another subsection?
Response: We are very sorry for our negligence of some details. This title was wrong here. It's been removed.
- Response to comment:
Other bulleted lists: Either change to numbered lists if appropriate, or I would suggest just writing these as logical paragraphs. Certainly, as mentioned above, the Conclusions should be written in paragraph structure.
Response: Thank you for your comment. We have made correction according to your comments.
- Response to comment:
The figures look okay to me, some oddities with text labels overlapping axis lines which can be fixed up if possible. Otherwise, these are quite good at conveying the key results.
Response: Your comment is very helpful. We have re-draw the figures according to your suggestion (figures 6-8).
Special thanks to you for your good comments.
We tried our best to improve the manuscript and made some changes in the expression of English. These changes will not influence the content and framework of the paper. And here we did not list the changes but marked in red in revised paper.
We appreciate for editors and reviewers’ warm work earnestly, and hope that the correction will meet with approval.
Once again, thank you very much for your comments and suggestions.

Reviewer 2 Report
Abstract: The abstract is not clear enough in terms of the significant of the results. The style can also be more concise and flow more logically. For example, the sentence "According to the simulation's results, the operation will be straightforward" is really vague. What about the operation will be straightforward?
Introduction: This is generally fine but is too lengthy and takes too long to get the the main thrust of the study.
Methods: This section is generally clear and detailed. There are some English language and formatting issues to be resolved that would improve readability.
Results: In Figure 6, the simulated seems to consistently underestimate the actual data. What is the reason for this in the discussion and the implications of this result for operations?
Author Response
Reviewer #2:
- Response to comment:
Abstract: The abstract is not clear enough in terms of the significant of the results. The style can also be more concise and flow more logically. For example, the sentence "According to the simulation's results, the operation will be straightforward" is really vague. What about the operation will be straightforward?
Response: We are sorry that we didn’t make ourselves clear. We meant “the operation will be convinient”. The abstract has been revised according to your comments.
- Response to comment:
Introduction: This is generally fine but is too lengthy and takes too long to get the main thrust of the study.
Response: Thank you for your comment. We have re-written the introduction according to the Reviewer’s suggestion.
- Response to comment:
Methods: This section is generally clear and detailed. There are some English language and formatting issues to be resolved that would improve readability.
Response: We are sorry for the English expression. The language has been modified.
- Response to comment:
Results: In Figure 6, the simulated seems to consistently underestimate the actual data. What is the reason for this in the discussion and the implications of this result for operations?
Response: We are very sorry for our negligence of the simulated result. These contents have been added in the discussion (lines 487-498).
The added part is as follows.
The simulated water level in Figure 6 is lower than the actual data for most of the time, because a larger discharge flow is adopted in the model than the actual one at the beginning, resulting in a decline of the simulated water level. In the middle part of the period, the simulated outflow discharge is close to the actual one, so the simulated water level is still persistently lower than the actual data. The simulated initial water level being low is beneficial for flood control, but may lead to both a lower water head and a less benefit of the power station. In order to improve the practicality of the simulation, the scheduling model can be further optimized. In order to be closer to the actual result, the total scheduling process was divided into several smaller periods, and the restriction conditions of each period was set through manual interaction [44, 45]. This is actually from the idea of mathematical programming method, which can be regarded as a direction of continuous improvement.
Special thanks to you for your good comments.
We tried our best to improve the manuscript and made some changes in the expression of English. These changes will not influence the content and framework of the paper. And here we did not list the changes but marked in red in revised paper.
We appreciate for editors and reviewers’ warm work earnestly, and hope that the correction will meet with approval.
Once again, thank you very much for your comments and suggestions.

Reviewer 3 Report
General comments
The title does not corresponding the content of the article. The content of article shows that it concerns the modeling of the discharge of a flood wave flowing through a reservoir, while considering the security of downstream flood management, which is not reflected in the title, relating to modeling of the water resource planning. Moreover it is difficult to clearly assess what the article is about, if it is about the programming in Python language the model of a flood wave passing through a reservoir, or it is about modeling the passage of the flood wave through a reservoir. Because the large part of article is about programming in Python.
The abstract does not reflect the content of the article as it mostly concerns the process of programming the model and the language programming. It is needs to be improved. Abstract – The aim, methods, results and conclusions. Must be noted generally and shortly .
The "Discussion" chapter basically relates to the programming of the model and the model itself, and there is no reference and discussion to the research results described in the results chapter. Moreover, there is no reference / comparison of the received research results, to the works and research results of other researchers. This should be completed in chapter "Discussion"
Conclusions should be a result from the research carried out in the article, obtained and discussed research results, but the conclusion no. 1 is not directly connected to the research results.
"References" should be to be extended.
Specific comments
Figure 1. Add a maps showing the location of the research area in the map of China, province and district
Figure 6-8 Why is the water level in "m3 / s" and not in "m"?
Author Response
Reviewer #3:
- Response to comment:
The title does not corresponding the content of the article. The content of article shows that it concerns the modeling of the discharge of a flood wave flowing through a reservoir, while considering the security of downstream flood management, which is not reflected in the title, relating to modeling of the water resource planning. Moreover it is difficult to clearly assess what the article is about, if it is about the programming in Python language the model of a flood wave passing through a reservoir, or it is about modeling the passage of the flood wave through a reservoir. Because the large part of article is about programming in Python.
Response: Thank you for your comment. We are sorry that we didn't make ourselves clear.
Considering your suggestion, we have revised the title from "Water Resource Scheduling" to "Integrated Water Resource Scheduling". In a broad sense, the concept of integrated water resources scheduling refers to the regulation and control of water resources through water conservancy projects, not only referring to the water resource management, but also the scheduling of reservoir projects aiming at flood control. As a result, we adopt the "integrated water resources scheduling model" as a comprehensive concept to describe flood control scheduling of reservoir as well as water distribution scheduling.
We propose a platform plug-in application model in this paper. Taking the integrated water resources scheduling as an example, the process of model plug-in construction and interface integration is introduced. Python language shows good advantages in the field of Web development and cloud computing, so it is chosen as the development language of the model.
In order to focus more on the platform plug-in approach, some text concerning the Python language have been deleted and is only reflected in lines 367-381, with an explanation of the choice of development language.
- Response to comment:
The abstract does not reflect the content of the article as it mostly concerns the process of programming the model and the language programming. It is needs to be improved. Abstract – The aim, methods, results and conclusions. Must be noted generally and shortly .
Response: Those comments are all valuable and very helpful for revising and improving the abstract. The abstract was re-organized according to the suggestions.
- Response to comment:
The "Discussion" chapter basically relates to the programming of the model and the model itself, and there is no reference and discussion to the research results described in the results chapter. Moreover, there is no reference / comparison of the received research results, to the works and research results of other researchers. This should be completed in chapter "Discussion"
Response: Those comments are important guiding significance to revise the discussion. We added the discussion of the simulation results as well as the outlook and suggestions for future application of model plug-in (lines 486-527).
- Response to comment:
Conclusions should be a result from the research carried out in the article, obtained and discussed research results, but the conclusion no. 1 is not directly connected to the research results.
Response: Thank you for your comments. We have made correction to the conclusions (lines 528-553) .
- Response to comment:
"References" should be to be extended.
Response: Thank you for your comment. We have extended the "References".
- Response to comment:
Specific comments
Figure 1. Add a maps showing the location of the research area in the map of China, province and district
Figure 6-8 Why is the water level in "m3 / s" and not in "m"?
Response: Thanks for the good comments. According to your suggestion, the corresponding map has been added, as shown in Figure 1.
We are very sorry for our incorrect writing of the units of water level in Figure 6-8, and the units have been corrected to “m”.
Special thanks to you for your good comments.
We tried our best to improve the manuscript and made some changes in the expression of English. These changes will not influence the content and framework of the paper. And here we did not list the changes but marked in red in revised paper.
We appreciate for editors and reviewers’ warm work earnestly, and hope that the correction will meet with approval.
Once again, thank you very much for your comments and suggestions.

Round 2
Reviewer 3 Report
The article is the authors' idea to present a certain issue. The authors are responsible for its content. If the authors believe that introduced corrections are sufficient and that further corrections will change the purpose and subject of the article, I will not insist on further corrections. Although I would describe the presented problem a little bit different.
I am not 100% satisfied with the corrections made by the authors to the article. However, I accept the article as an author's implementation of the research idea and presentation of its results.